# Comparison of the Effects of Brazil Nut Oil and Soybean Oil on the Cardiometabolic Parameters of Patients with Metabolic Syndrome: A Randomized Trial

**DOI:** 10.3390/nu12010046

**Published:** 2019-12-23

**Authors:** Lívia Martins Costa e Silva, Maria Luisa Pereira de Melo, Fernando Vinicius Faro Reis, Marta Chagas Monteiro, Savio Monteiro dos Santos, Bruno Alexandre Quadros Gomes, Luiza Helena Meller da Silva

**Affiliations:** 1LAMEFI—Laboratory of Physical Measures, Postgraduate Program in Food Science and Technology, Federal University of Para, Belém Pará 66075-900, Brazil; 2Academic Master’s Degree in Nutrition and Health, State University of Ceará, Fortaleza 60.714.903, Brazil; 3Institute of Health Sciences, Federal University of Para, Belém Pará 66075-900, Brazil; 4Postgraduate Program in Pharmaceutical Sciences, Federal University of Para, Belém Pará 66075-900, Brazil; 5Neuroscience and Cellular Biology Postgraduation Program, Biological Science Institute, Federal University of Para, Belém Pará 66075-900, Brazil

**Keywords:** metabolic syndrome, cardiovascular diseases, Brazil nuts, plant oils, *Bertholletia*

## Abstract

Recent evidence suggests that replacing saturated fat with unsaturated fat is beneficial for cardiovascular health. This study compared the effects of Brazil nut oil (BNO) and soybean oil (SO) supplementation for 30 days on anthropometric, blood pressure, biochemical, and oxidative parameters in patients with metabolic syndrome (MS). Thirty-one patients with MS were randomly allocated to receive 30 sachets with 10 mL each of either BNO (*n* = 15) or SO (*n* = 16) for daily supplementation. Variables were measured at the beginning of the study and after 30 days of intervention. No change in anthropometric and blood pressure variables were observed (*p* > 0.05). Total (*p* = 0.0253) and low-density lipoprotein (*p* = 0.0437) cholesterol increased in the SO group. High-density lipoprotein cholesterol decreased (*p* = 0.0087) and triglycerides increased (*p* = 0.0045) in the BNO group. Malondialdehyde levels decreased in the BNO group (*p* = 0.0296) and total antioxidant capacity improved in the SO group (*p* = 0.0110). Although the addition of oils without lifestyle interventions did not affect anthropometric findings or blood pressure and promoted undesirable results in the lipid profile in both groups, daily supplementation of BNO for 30 days decreased lipid peroxidation, contributing to oxidative stress reduction.

## 1. Introduction

Metabolic syndrome (MS) includes the combination of at least three cardiovascular risk factors, such as obesity, dyslipidemia, hypertension, and hyperglycemia or type 2 diabetes, which are associated with insulin resistance, oxidative stress, and inflammatory status [1]. MS is highly prevalent in countries with a high rate of obesity and Western eating patterns, with consequential increases in health costs [2].

There is currently no effective recommendation for the prevention of MS other than lifestyle-based interventions aimed at normalizing body weight and maintaining good cardiometabolic control, including the maintenance of adequate levels of serum lipids, glucose, and blood pressure [2].

Recent evidence suggests that some nutrients, foods, and dietary patterns have positive effects on the management of MS, although there is still no consensus on which strategy is best [3,4,5]. However, most of the guidelines suggest that diet plays a fundamental role in the prevention and treatment of MS. Food guidelines for macronutrient intake and cardiovascular health generally emphasize the consumption of foods rich in monounsaturated fatty acids (MUFAs) and a reduction in saturated fatty acid (SFA) intake [6,7,8,9].

Given the current evidence on the role of food in the prevention of nutrition-related diseases, foods with biologically active ingredients that contribute to the prevention of these diseases are increasingly sought [10,11]. The biodiversity of the Amazon provides many plants that are rich in bioactive compounds. The Brazil nut (*Bertholletia excelsa*), one of the most important oilseeds from this region, has an internationally known nutritional richness [12]. Bioactive substances present in Brazil nuts have already been identified [13] along with their proven beneficial anti-inflammatory [14] and endothelial effects [15]. This oleaginous plant has a complex matrix, rich in bioactive substances, such as selenium, α- and γ-tocopherols, phenolic compounds, folate, magnesium, calcium, proteins, MUFAs, and polyunsaturated fatty acids (PUFAs) [14]. The high content of MUFAs and PUFAs in foodstuffs is potentially beneficial to cardiovascular health, especially when compared to diets rich in SFAs [16].

The oil from the Brazil nut has a high nutritional value, including an unsaturated fatty acid (UFA) content of approximately 75%, constituted predominantly of oleic and linoleic acids, and phytosterols and vitamins A and E [17].

Considering the current evidence on the effects of UFA intake on cardiovascular health, this study aimed to compare the effects of Brazil nut oil and soybean oil supplementation for 30 days on anthropometric, blood pressure, biochemical, and oxidative parameters in patients with MS.

## 2. Materials and Methods

### 2.1. Participants

The participants were adults aged between 36 and 65 years, of both genders, diagnosed with MS according to the criteria of the National Cholesterol Education Program Adult Treatment Panel III (NCEP-ATP III, 2001) [18]. All participants were recruited from the Nutrition Clinic of the João de Barros Barreto University Hospital (HUJBB), located in the city of Belém, Pará, Brazil. Individuals with thyroid function alterations, neurodegenerative, or chronic liver diseases and smokers were excluded from the study.

All participants were duly informed of the procedures used and gave their written consent. Data collection was carried out between September 2017 and October 2017. The research was approved by the Research Ethics Committee of the HUJBB (protocol No. 68270517.0.0000.0017) and registered at Brazilian Clinical Trials Registry (RBR-5d25w3).

### 2.2. Experimental Design

A randomized, double-blind, placebo-controlled clinical trial was conducted on 41 patients with MS. The participants were randomly separated into two groups through the distribution of unidentified sachets. This allowed the formation of the experimental group, which received extra-virgin Brazil nut oil (BNO, *n* = 20), and the control group, which received soybean oil (SO, *n* = 21). SO was chosen as the placebo, considering that it is the most used by the Brazilian population in culinary preparations. Neither the participant nor the investigators (responsible for data collection, data analysis, and oil delivery) knew to which group the research participants belonged. A flow chart representing the experimental protocol is presented in Figure 1.

Volunteers from both groups received 300 mL of the oils, stored in identical 10 mL sachets of opaque and non-toxic material, which were previously randomly coded; only at the end of the analysis were the codes revealed. Each participant was instructed to use one sachet for lunch, without heating, once a day, for 30 consecutive days (10 mL/day).

The lifestyle habits of each subject were assessed through an interview and clinical consultation. Subjects were guided to maintain their eating habits and lifestyle, and to avoid food supplements.

All evaluations were performed at the beginning of the study (T0) and after 30 days of intervention (T1).

### 2.3. Oils

Brazil nut oil was obtained from an industry specialized in the production of Amazonian oils, located in Tapanã, Pará, Brazil, while soybean oil was purchased from a supermarket in Belém, Pará, Brazil.

#### Determination of the Fatty Acid Profile of Brazil Nut Oil

The composition of fatty acids was determined by converting fatty acids into methyl esters (FAMEs) based on the method proposed by Rodrigues et al. [19]. The fatty composition was then examined using gas chromatography (CP3380) equipped with a flame ionization detector and with a CP-Sil 88 capillary column (60 m in length, with an internal diameter of 0.25 mm and a thickness of 0.25 mm; Varian Inc., CA, USA). The operating conditions included helium as a carrier gas, with a flow rate of 0.9 mL/min, a detector (FD) at 250 °C, an injector (split ratio of 1:100) at 245 °C, and an injection volume of 1 μL. Program temperature of the column occurred over 4 min at 80 °C and a subsequent increase to 220 °C at 4 °C/min. Individual fatty acid peaks were identified by comparing the obtained retention times with the retention times of known standard fatty acid mixtures (74X, Nu-check-Prep., Inc., MN, USA), examined under the same operating conditions.

### 2.4. Assessments

#### 2.4.1. Body Composition and Blood Pressure Parameters

The same trained evaluator collected the anthropometric measurements (weight, height, and waist circumference), as described by Lohman, Roche and Martorell [20] and as recommended by the World Health Organization [21]. Percent body fat (%BF) was obtained for each participant through bioimpedance.

All participants were evaluated in the morning. Body mass index (BMI) was calculated by measuring weight and height and dividing the weight in kilograms (kg) by the square of height in meters (m). Weight was measured with patients barefoot and wearing light clothing, and height was determined using a portable stadiometer.

Waist circumference (WC) was measured using an inelastic and flexible measuring tape with a scale ranging from 0 to 150 cm and resolution of 0.1 cm.

Systolic blood pressure (SBP) and diastolic blood pressure (DBP) at rest were measured with the arm positioned and supported at the level of the heart and the palm of the hand facing upwards, according to the procedures described by Veiga et al. [22]. Three measurements were taken and the results of these were used to determine the participant’s mean SBP and DBP.

#### 2.4.2. Biochemical Parameters

Participants fasted for 12 h before their blood samples were collected by venipuncture. All samples were centrifuged at 3000 rpm for 15 min, and the plasma and serum obtained were stored at −70 °C until analysis was performed.

Fasting blood glucose, total cholesterol, high-density lipoprotein (HDL) cholesterol and triglycerides were analyzed in a biochemical self-analyzer (Dade AR^®^) using Dade Behring kits. Low-density lipoprotein (LDL) cholesterol was determined according to the Friedwald equation [23].

#### 2.4.3. Oxidative Parameters

The total antioxidant capacity was determined using the Trolox equivalent antioxidant capacity (TEAC) assay. Trolox (6-hydroxy-2,5,7,8-tetramethyl-croman-2-carboxylic acid; SigmaAldrich 23881-3) is a powerful water-soluble antioxidant analog to vitamin E. We used the method proposed by Miller et al. [24] and modified by Re et al. [25], which is a colorimetric technique based on the reaction between ABTS (SigmaAldrich A1888) and potassium persulfate (K2S2O8; Sigma-Aldrich 60490), resulting in the production of ABTS+• radical cation (radical 2,2-azinobis [3-ethylbenzothiazoline-6-sulfonate], diammonium salt), green/blue chromophore [26]. The addition of antioxidants (present in the sample) to this preformed cation reduces ABTS proportionally to the antioxidant capacity, the concentration of antioxidants, and the duration of the reaction. This process was measured by spectrophotometry by observing the change in absorbance, read at 734 nm for 5 min. The results were expressed in μM/mL.

Lipid peroxidation was determined using the method originally created by Khonn and Liversedge [27] and adapted by Percário et al. [28]. It is a technique based on the reaction of malondialdehyde (MDA) with thiobarbituric acid (TBA; Sigma-Aldrich T5500) at pH 2.5 at 94 °C, forming a pink MDA–TBA complex, with absorption at 535 nm. Since the reaction is not specific for MDA, because TBA can react with sugars, amino acids, proteins, and bilirubin, the term thiobarbituric acid-reactive substances (TBARSs) is used [29]. TBARS levels were expressed as malondialdehyde (MDA) equivalent.

The technical procedure consisted of an initial preparation of monobasic potassium phosphate (KH2PO4 75 mM, Synth, 35210) in acidified water (pH 2.5). This solution was used in the preparation of TBA (10 nM). One hundred microliters of the sample was added to 500 µL of TBA 10 nM solution. It was then water bath-heated (94 °C for 60 min). After incubation, the solution was cooled to room temperature for 10 min. Then, 2.0 mL of 1-butyl alcohol was added and it was vigorously homogenized in a vortex and later subjected to centrifugation at 2500 rpm for 10 min before 1.0 mL of supernatant was collected for spectrophotometric reading at 535 nm. The MDA standard (1,1,3,3, tetrahydroxypropane—Sigma-Aldrich, T9889) was used to generate the standard curve, and the results were expressed in μM/L.

### 2.5. Statistical Analysis

Descriptive and inferential statistics were used to evaluate the difference between T0 and T1 in the two groups (BNO and SO). Quantitative variables were presented by central tendency and variation measures. The Shapiro–Wilk test was used to evaluate the distribution of quantitative variables. Gender was compared using Fisher’s exact test. In order to evaluate the difference between the groups, we used the Student’s t-test (for normal samples) and the Mann–Whitney U-test (when non-normality was detected in the sample). To evaluate intragroup differences (T0 × T1), either the Student’s *t*-test or the Wilcoxon Signed-rank test was used for paired samples for data with normal and non-normal distributions, respectively. The Alpha Error = 0.05 (5%) was previously set for null hypothesis rejection. Statistical processing was performed using the software BioEstat version 5.3 [30].

## 3. Results

The fatty acid profiles of Brazil nut and soybean oils are described in Table 1. Both are rich in unsaturated fatty acids (BNO: 72.89% and SO: 84.47%); oleic MUFA w9 predominates in BNO (37.58%), while linoleic PUFA w6 predominates in SO (54.6%).

The total sample was composed of 31 individuals, divided into two groups (BNO *n* = 15 and SO *n* = 16). The distribution by gender did not show any significant difference between the groups (*p* > 0.5), and the two groups were mostly composed of women (BNO: 73.3% and SO: 93.7%). Similarly, age did not show any significant difference between the groups (*p* > 0.5), ranging from 36 to 64 years in the BNO group (mean 58.5 ± 8.4) and from 43 to 65 years in the SO group (mean 58.3 ± 5.7 years).

The anthropometric and blood pressure data are described in Table 2. There were no significant differences in BMI, %BF, WC, SBP, and DBP values between the groups at the beginning of the study (T0) and at the end of the intervention (T1). Similarly, there was no difference in BMI, %BF, WC, SBP, and DBP between the groups before and after the use of oils (T1 − T0).

The biochemical data are shown in Table 3. Regarding the lipid profile, the SO group showed an increase in total cholesterol at the end of the intervention (T1 − T0) (*p*-value = 0.0253) and in LDL cholesterol values (*p*-value = 0.0437). The BNO group, in turn, showed a decrease in HDL cholesterol (T1 − T0) (*p*-value = 0.0087) and an increase (*p* = 0.0045) in triglycerides. There was no difference between the glycemic values of the groups at the beginning of the study (T0) (*p*-value = 0.2582); after 30 days of intervention, the BNO group showed significantly higher levels of glucose (*p* = 0.0282) and the variation in glucose between the groups (T1 − T0) was significant (*p* = 0.0368), although glycemic levels intragroup did not significantly change at the end of the supplementation (T1 − T0) (*p*-value > 0.05).

Table 4 shows the biomarkers of oxidative metabolism before and after 30 days of supplementation. After the intervention, MDA levels were significantly lower (*p*-value = 0.0296) in the group that received Brazil nut oil. The soybean group showed statistically significant improvement in TEAC (*p*-value = 0.0110), suggesting an improvement in antioxidant capacity (Figure 1).

Study data are available in Appendix A.

## 4. Discussion

To the best of our knowledge, this is the first study to compare the effects of 10 mL supplementation of Brazil nut and soybean oils in patients with MS.

Brazil nut and soybean oils are rich in UFAs, with values of 72.89% and 84.47% [31], respectively. Oleic acid w9 predominates in BNO (34.71%) and linoleic acid w6 predominates in SO (56.6%) [31]. These results are consistent with what has been reported in the literature [17,32,33].

The influence of the type of fatty acid ingested on CVD risk factors and on plasma concentrations of lipids and lipoprotein has been demonstrated for a long time in several experimental and population studies [34,35,36].

The results of our clinical trial showed that supplementation of 10 mL of Brazil nut and soybean oils for 30 days did not promote significant changes in anthropometric and blood pressure parameters. Similar results were found in other studies with approximate intervention periods [37,38]. Longer intervention periods are needed in order to determine whether this supplementation could have a more significant impact on these parameters [39,40,41,42].

Many mechanisms suggest that the PUFAs have effects on the modulation of pathways that could influence lipid profile. For example, PUFAs can decrease the synthesis of very low-density lipoprotein (VLDL), both due to the greater catabolism of these fatty acid in peroxisomes and its interference with nuclear receptors. Other effects include the increase in the fluidity of cell membranes of hepatocytes, interfering with the activity of LDL receptors [43] and with the amount of hepatic apoprotein B/E receptors [44]. In addition, PUFAs can promote changes in the LDL spatial configuration, providing a lower volume available for the incorporation and transport of cholesterol [45], besides decreasing triglyceridemia due to the stimulation of the hydrolysis of apo B-100 [46].

Oleic acid (MUFA), in turn, helps in reducing LDL plasma levels without oxidizing it [47], probably because it is a better substrate for acyl-coenzyme A:cholesterol acyltransferase (ACAT) in the liver, and therefore, free cholesterol is rapidly esterified, not leading to the suppression of LDL receptors [48]. Moreover, this fatty acid induces less endogenous synthesis of cholesterol than other types of fatty acid [49].

It is important to note that the intake of SFAs and total fat cannot be neglected as their high consumption can promote changes in glucose homeostasis, high blood pressure, and hypertriglyceridemia, regardless of the amount of MUFAs and PUFAs ingested [50].

In our trial, the BNO group showed an increase in triglyceride levels and a decrease in HDL levels, while the control group (SO) showed an increase in total cholesterol and LDL levels. Both groups were mostly composed of women with median ages of 58.5 ± 8.4 (BNO group) and 58.3 ± 5.7 (SO group) years; this is the age at which many women are in menopause. It is known that menopause may contribute to lipid profile alteration [51], but in our study, this effect was not evaluated. Studies evaluating the effects of oil supplementation in patients with chronic diseases present divergent data on lipid profiles; Harris, Hutchins and Fryda [38] showed negative results [38], while others found improvements in these parameters [52,53,54]. These differences among studies may be explained by the different nutritional contexts in which the studies were performed, in addition to the dosage and the period of use of the oils.

Although recommendations for the treatment of MS usually include the replacement of saturated fat and monosaccharide by unsaturated fats, it is not yet clear which metabolic parameters are improved when saturated fat is replaced by higher concentrations of MUFAs or PUFAs [42]. Some studies that replaced SFAs by UFAs also obtained inconsistent results in cardiometabolic risk factors [55,56]. Moreover, other factors may lead to increased serum lipids and blood glucose, such as a low-fiber diet, a high intake of refined carbohydrates and saturated fat, and a sedentary lifestyle [51]. According to Astrup [57], the replacement of saturated fat by polyunsaturated fat present in vegetable oils may even increase the risk of CVD death if they are not from the omega-3 series.

Supplementation of the two types of oils (SO and BNO) did not significantly change the glucose levels, which is consistent with previous studies [42,58,59].

In our study, oxidative metabolism was evaluated by measuring lipid peroxidation products, such as MDA, and the total antioxidant capacity in plasma. The analysis of the total antioxidant capacity is recommended more than the analysis of isolated antioxidants, due to their interaction in plasma or serum and because the total antioxidant capacity represents the cumulative action of all antioxidants present [26].

The decrease in MDA levels observed after the use of Brazil nut oil thus suggests a reduction in lipid peroxidation, showing a desirable effect of this oil in the treatment of MS. This can be evidenced by the reduction in the MDA/triglycerides ratio (approximately 50%) (Table 4) and allows us to conclude that, despite the increase in triglycerides, Brazil nut oil was able to decrease this parameter.

The increase in total antioxidant capacity detected in the SO group may be associated with the tocopherol contents present in this oil [31]. In contrast, it may be related to the observed increase in total cholesterol and LDL, indicating a probable defense response of the body against vascular oxidative stress [60]. This response occurs through a negative feedback loop that can activate enzymatic pathways of the antioxidant defense system in order to reduce intracellular ROS levels and thereby minimize oxidative damage [61].

The effect on oxidative stress depends on the type and total amount of macronutrients consumed [62]. These two aspects may contribute to the oxidative stress and favor the development of obesity and chronic diseases [63]. The molecular mechanisms of nutrient-mediated oxidative stress, however, are complex and still not well-known [64].

Several studies indicate that when polyphenols are present in the diet, they hinder the development of disorders related to oxidative stress, such as cancer, cardiovascular diseases, diabetes mellitus, and MS [65]. These compounds are commonly found in products of plant origin and represent the most abundant sources of antioxidants in our diet [66].

Vegetable oils are sources of bioactive compounds such as tocopherols and phenolic compounds. The average tocopherol content in Brazil nut and soybean oils is approximately 190 µg/g and 980 µg/g, respectively, while for phenolic compounds, these values are 150 mg gallic acid equivalent (GAE)/100 g for BNO and 6 mg GAE/100 g for SO [31,67,68,69,70,71]. Despite containing lower tocopherol levels and SO [31], this amount still provides antioxidant effects [68]. Moreover, its content of phenolic compounds reaffirms the antioxidant power of BNO [72], which probably determined its effect on lipid peroxidation.

Given these results, it can be observed that both oils are sources of antioxidants. Soybean oil is rich in tocopherols and Brazil nut oil in phenolic compounds. The performance of antioxidants, however, depends on several factors (e.g., the type and location of the generated free radicals and the optimal doses). In this sense, it is possible for an antioxidant to act as a protector in one system but fail to protect or even increase induced damage to other tissues [73]. Furthermore, evidence suggests that in addition to phenolic compounds, MUFAs present in Brazil nuts improve oxidative stress [74].

This study shows that the potential benefits of BNO for the treatment of MS are limited without an overall healthy lifestyle.

## 5. Conclusions

Although the addition of oils without lifestyle interventions did not alter the anthropometric and blood pressure parameters and promoted undesirable results in the lipid profile, daily supplementation of 10 mL of Brazil nut oil for 30 days decreased lipid peroxidation, contributing to oxidative stress reduction.

## Figures and Tables

**Figure 1 nutrients-12-00046-f001:**
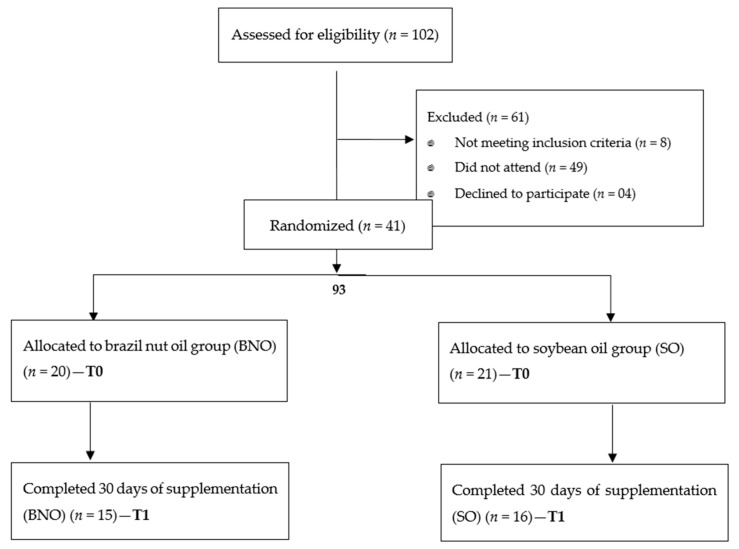
Consolidated Standards of Reporting Trials (CONSORT) diagram of participants representing the experimental groups.

**Table 1 nutrients-12-00046-t001:** Profile of fatty acids from Brazil nut oil and soybean oil.

Fatty Acid	Usual Name	Brazil Nut Oil (%)	Soybean Oil * (%)
C8:0	Caprylic Acid	0.03	-
C10:0	Capric Acid	0.04	-
C12:0	Lauric Acid	1.19	-
C14:0	Myristic Acid	0.72	-
C16:0	Palmitic Acid	14.89	11.50
C16:1	Palmitoleic Acid	0.32	-
C17:0	Margaric Acid	0.07	-
C18:0	Stearic Acid	10.04	3.62
C18:1	Oleic Acid	37.58	24.0
C18:2	Linoleic Acid	34.71	54.6
C18:3	α-Linolenic Acid	0.28	5.87
C20:0	Arachidic Acid	0.08	0.46
C22:0	Behenic Acid	0.05	-
∑ Saturated		27.11	15.58
∑ Monounsaturated		37.90	24.0
∑ Polyunsaturated		34.99	60.47

* Source: Zaunschirm et al. [31].

**Table 2 nutrients-12-00046-t002:** Anthropometric and blood pressure characteristics of individuals with metabolic syndrome (MS) before and after supplementation with Brazil nut oil or soybean oil.

	BNO			SO		
	Before	After	Δ	Before	After	Δ
Sample size	15	15	15	16	16	16
BMI	32.2	31.5	−0.7	30.8	30.8	−0.1
	±4.4	±4.4	1.2	±4.4	±4.5	0.3
*p*-value (intragroup)			0.0566			0.4539
	T0	T1	Δ (T1 − T0)			
*p*-value (BNO × SO)	0.4125	0.6601	0.1280			
%BF	39.9	40.3	0.4	43.7	44.3	0.6
	±8.1	±8.7	1.4	±6.6	±6.4	1.4
*p*-value (intragroup)			0.9089			0.8045
	T0	T0	Δ (T1 − T0)			
*p*-value (BNO × SO)	0.0782	0.0904	0.8712			
WC	109.1	108.2	−1.3	102.3	101.3	−1.0
	±10.3	±8.8	2.7	±10.7	±9.8	3.6
*p*-value (intragroup)			0.3224			0.2859
	T0	T1	Δ (T1 − T0)			
*p*-value (BNO × SO)	0.1303	0.0939	0.9569			
SBP	132.0	130.6	−1.4	133.9	125.8	−8.2
	±19.8	±16.7	12.7	±17.3	±14.4	16.6
*p*-value (intragroup)			0.5434			0.4456
	T0	T1	Δ (T1 − T0)			
*p*-value (BNO × SO)	0.7824	0.4073	0.3705			
DBP	78.5	78.2	−0.3	78.8	76.3	−2.5
	±11.8	±10.4	4.1	±6.8	±11.0	10.2
*p*-value (intragroup)			0.8062			0.3051
	T0	T1	Δ (T1 − T0)			
*p*-value (BNO × SO)	0.5532	0.4768	0.5673			

BNO, Brazil nut oil; SO, soybean oil; BMI, body mass index; %BF, percent body fat; WC, waist circumference; SBP, systolic blood pressure; DBP, diastolic blood pressure. *p*-value (BNO × SO) by Student’s *t*-test (IMC, WC, SBP, and DBP). Data are expressed as the mean ± standard deviation.

**Table 3 nutrients-12-00046-t003:** Biochemical characteristics of individuals with MS before and after supplementation with Brazil nut oil or soybean oil.

	BNO			SO		
	Before	After	Δ	Before	After	Δ
Sample size	15	15	16	16	16	16
Cholesterol	243.1	240.0	−3.1	216.6	240.5	23.9
	±59.9	±53.4	±66.2	±35.2	±38.8	±38.6
*p*-value (intragroup)			0.8604			0.0253 *
	T0	T1	Δ (T1 − T0)			
*p*-value (BNO × SO)	0.3738	0.9370	0.2684			
LDL	161.3	151.3	−9.9	132.6	155.3	22.7
	±58.8	±59.0	±68.6	±37.5	±38.7	±43.6
*p*-value (intragroup)			0.5823			0.0437 *
	T0	T1	Δ (T1 − T0)			
*p*-value (BNO × SO)	0.2599	0.9656	0.2434			
HDL	36.5	29.7	−6.8	40.2	42.6	2.2
	±8.1	±8.2	±8.7	±12.2	±14.3	±9.9
*p*-value (intragroup)			0.0087 *			0.3499
	T0	T1	Δ (T1 = T0)			
*p*-value (BNO × SO)	0.2948	0.1665	0.0121 *			
Triglycerides	226.7	295.3	68.6	218.3	213.8	−4.5
	±32.1	±73.8	±83.6	±36.6	±46.0	±−4.5
*p*-value (intragroup)			0.0045 *b			0.8501
	T0	T1	Δ (T1 − T0)			
*p*-value (BNO × SO)	0.8451	0.0021 *a	0.0066 *a			
Glucose	153.4	157.6	4.1	115,8	106.8	−9.0
	±78.1	±68.2	±43.1	±32	±26.9	±38.8
*p*-value (intragroup)			0.7143			0.3678
	T0	T1	Δ (T1 − T0)			
*p*-value (BNO × SO)	0.5016	0.0282 *c	0.0368 *			

BNO, Brazil nut oil; SO, soybean oil; TC, total cholesterol; LDL, low-density lipoprotein; HDL, high-density lipoprotein; TG, triglycerides. * (Before × After): Student’s *t*-test for paired samples (TC, LDL, HDL). a* Comparison (BNO × SO): the Mann–Whitney U-test (TG). b* Intragroup comparison by Wilcoxon Signed-Rank test. (TG). c* The Mann–Whitney U-test for paired samples (glucose). Data are expressed as the mean ± standard deviation.

**Table 4 nutrients-12-00046-t004:** Markers of the oxidative metabolism of individuals with MS before and after supplementation with Brazil nut oil or soybean oil.

	BNO			SO		
	Before	After	Δ	Before	After	Δ
Sample size	14	14	14	15	15	15
TEAC	3.20	3.26	0.06	3.39	3.50	0.10
	±0.80	±0.69	±1.12	±0.10	±0.10	±0.13
*p*-value (intragroup)			0.8727			0.0110 *
	T0	T1	Δ (T1 − T0)			
*p*-value (BNO × SO)	0.3041	0.4767	0.6112			
MDA	3.18	2.27	−0.84	2.91	2.60	−0.31
	±0.93	±0.96	±1.92	±0.48	±1.42	±1.51
*p*-value (intragroup)			0.0296 *			0.4511
	T0	T1	Δ (T1 − T0)			
*p*-value (BNO × SO)	0.9658	0.8362	0.5053			
MDA/triglycerides (nmol/mg)	1.40	0.77	−0.63	1.33	1.22	−0.12
MDA/TEAC	0.99	0.69	−0.3	0.86	0.74	−0.12

* Student’s *t*-test for paired samples. Data are expressed as the mean ± standard deviation. TEAC, Trolox equivalent antioxidant capacity; MDA, malondialdehyde; T0, beginning of the study; T1, after 30 days of intervention.

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
