# Peer review of "Comparison of the Effects of Brazil Nut Oil and Soybean Oil on the Cardiometabolic Parameters of Patients with Metabolic Syndrome: A Randomized Trial"

_nutrients, 2019, doi:10.3390/nu12010046_

Round 1

Reviewer 1 Report

Nutrients
Original Article 620585 titled “Comparison of the Effects of Brazil Nut Oil and
Soybean Oil on the Cardiometabolic Parameters of Patients with Metabolic
Syndrome: A Randomized Trial." by Costa e Silva et al.
The present article is appropriate to this Journal, nevertheless, a considerable number
of major points raised by this reviewer should be clarified.

MAJOR POINTS
Results
1. The profile of fatty acids from the soybean oil used in the present study should be specified in detail in the Table 1.
2. The authors underline that the two groups were mainly composed by women.
The age of these women range from 36 to 64 years in BNO group and 43 to 65 years in SO group. It is well known that plasma estrogen levels affects lipid profile and some of these women might be in the menopause age. This Referee is concerned by how the age of women could be affecting the differences in the results found in plasma lipids between the two groups.
3. The data of insulin and ISI (insulin sensitivity index) might be more relevant that only those of glycemia in Table 3.
4. Since BNO oil produces a significant diminution in HDL cholesterol and a
significant increase in triglycerides, without significant changes in LDL or total cholesterol, the utility of BNO in the treatment of MS results quite limited.
5. Since BNO oil produces an increase in plasma triglycerides, in order to
reinforce the effect of BNO for reducing lipid peroxidation, the data of TBARS
should be expressed as nmol of MDA/mg of triglycerides in Table 4.
Discussion
1. Since BNO and SO oils present similar percentages of UFA (around 70-75 %), the authors should remark that the beneficial effect of BNO for decreasing lipid peroxidation is not due to the unsaturated fatty acids.
2. The data for the content of tocopherol and phenolic compounds in BNO oil are provided by the authors, however these data for SO oil are lacking. If any difference, the beneficial effect of BNO decreasing lipid peroxidation would be possibly due to the tocopherols and phenolic compounds.
3. The Discussion section is too long and, in some way, chaotic and inconsistent. It should be considerably shortened. A better organization of this section would be quite useful for a better understanding.

MINOR POINTS
Results
1. Figures 2 and 3 should be deleted because they show the same data as Tables 3 and 4, respectively.

Reviewer 2 Report

The authors compared the effects of daily supplementation by either Brazil nut oil (BNO) (N=15) or soybean oil (SO)(N=16), for 30 days, on anthropometric, blood pressure, biochemical, and oxidative parameters, in 31 patients with Metabolic Syndrome (MS). They found that the interventions did not affect anthropometric and blood pressure variables, but had an undesirable impact on lipid profile; total- and LDL-cholesterol increased in the SO group; HDL-cholesterol decreased whereas triglycerides levels increased in the BNO group. Interestingly, lipid peroxidation –assessed by thiobarbituric acid-reactive substances- decreased in the BNO group, and total antioxidant capacity improved in the SO group. It was suggested that daily supplementation of BNO for 30 days may help in the treatment of MS.

This is an interesting area of research and also the manuscript is well written. However, I am wondering why participants who received SO composed the control (placebo) group? A true placebo group would be more appropriate. Moreover, the number of participants is small. What is the power of the study to detect differences between pre- and post-intervention? Besides, the period of intervention was short, as the authors have already mentioned (line 284).

All participants received extra BNO or SO; PUFAs did not replace saturated fats. What is the difference in omega-3 PUFAs between BNO and SO?

Are there any data on nutrition/diet of the study groups before and after intervention?

Minor comments:

In introduction, the association between MS and obesity is repeatedly presented (lines 43-44). Please revise

Line 288: showed an increase or decrease?

Figure 1. How many individuals declined to participate? 102 (assessed)-49 (excluded)=53 participants. Why only 41 were randomized?

How were participants allocated to each group?

Author Response

Dear Reviwer,

Comments and suggestions from reviewers are welcome, we hope to have answered and clarified all points of the previous review. We believe the paper is better and qualified to be published in Nutrients. We are looking forward to the experience of shared work and with the opportunity to continue with future contributions.

Thank you.

Round 2

Reviewer 1 Report

Recommendations and suggestions have been implemented in sufficient manner into the revised version of manuscript.

Reviewer 2 Report

The authors responded satisfactory to the comments and the manuscript has been improved.